# Clinical Impact of the Fracture Risk Assessment Tool on the Treatment Decision for Osteoporosis in Patients with Knee Osteoarthritis: A Multicenter Comparative Study of the Fracture Risk Assessment Tool and World Health Organization Criteria

**DOI:** 10.3390/jcm8070918

**Published:** 2019-06-26

**Authors:** Bo Young Kim, Hyoun-Ah Kim, Ju-Yang Jung, Sang Tae Choi, Ji-Min Kim, Sang Hyon Kim, Seong-Ryul Kwon, Chang-Hee Suh, Sung-Soo Kim

**Affiliations:** 1Division of Rheumatology, Department of Internal Medicine, Gangneung Asan Hospital, University of Ulsan College of Medicine, Gangneung 25440, Korea; 2Department of Rheumatology, Ajou University School of Medicine, Suwon 16449, Korea; 3Division of Rheumatology, Department of Internal Medicine, Chung-Ang University College of Medicine, Seoul 06973, Korea; 4Division of Rheumatology, Department of Internal Medicine, Keimyung University School of Medicine, Dongsan Medical Center, Daegu 42601, Korea; 5Division of Rheumatology, Department of Internal Medicine, Rheumatism Center, Inha University Hospital, Incheon 22332, Korea

**Keywords:** osteoporosis, fracture, fracture risk assessment tool, knee osteoarthritis

## Abstract

Background: To compare the frequency of high-risk osteoporotic fracture in patients with knee OA (OA) using the fracture risk assessment tool (FRAX) and the bone mineral density (BMD). Methods: We retrospectively assessed 282 Korean patients with knee OA who visited five medical centers and 1165 healthy controls (HCs) aged ≥50 years without knee OA. After matching for age, sex, and body mass index, 478 subjects (239 patients with knee OA and 239 HCs) were included. Results: Based on the BMD, the frequency of osteoporosis was 40.2% in patients with knee OA and 36.4% in HCs. The predicted mean FRAX major osteoporotic fracture probabilities calculated with or without femur neck BMD differed significantly between the knee OA and HCs (6.9 ± 3.8% versus 6.1 ± 2.8%, *p* = 0.000 and 8 ± 3.6% versus 6.8 ± 2.3%, *p* < 0.001, respectively). The mean FRAX hip fracture probabilities calculated with or without femur neck BMD differed significantly in the knee OA and HCs (2.1 ± 2.4% versus 1.7 ± 1.8%, *p* = 0.006 and 3 ± 2.3% versus 2.4 ± 1.6%, *p* < 0.001, respectively). Conclusion: Our study suggests that FRAX may have a clinical impact on treatment decisions to reduce osteoporotic facture in patients with knee OA.

## 1. Introduction

Osteoporosis and osteoarthritis (OA) are common bone disorders related to aging and are associated with significant morbidity and disability. The relationship between osteoporosis and OA is complex and controversial. Several previous studies have indicated an inverse relationship between osteoporosis and OA [1,2], results of which have suggested a protective effect of OA on osteoporosis. However, other studies suggest that increased bone mineral density (BMD) in OA does not reduce the risk of fracture [3,4]. Data from the Rotterdam study indicated that, although patients with knee OA had a higher BMD, their incident fracture risk was higher than that in those without knee OA [5]. The absence of a protective effect of increased BMD on fracture risk in these studies was explained as an increased tendency for falls in patients with OA. A recent study reported that inflammation contributes to the development of OA and osteoporosis, providing insight regarding the concomitant presence of both conditions [6]. Subsequent studies have reported that 20–29% of patients with advanced OA have occult osteoporosis [7,8,9]. Therefore, fracture risk assessment in patients with OA should not be overlooked.

The World Health Organization (WHO) criteria, using BMD measured by dual-energy X-ray absorptiometry (DXA), are the most widely used in the diagnosis of osteoporosis [10]. Although a low BMD is a major determinant of osteoporotic fracture, over 50% of fracture cases did not have a BMD considered osteoporosis [11,12]. The fracture risk assessment tool (FRAX) [13] is the most commonly used model for fracture risk assessment. The clinical risk factors assessed using the FRAX have high validity from an evidence-based assessment and are easily obtainable [14].

In clinical settings, physicians mainly determine osteoporosis treatment based on WHO criteria. However, whether these current treatments will be appropriate for patients with OA has not been clearly studied. Inappropriate approaches to fracture risk assessment may lead to under-treatment of osteoporosis. Thus, our aim was to evaluate the frequency of high-risk group of osteoporotic fracture in patients with knee OA comparing the FRAX and WHO criteria. We also examined whether patients with knee OA differ from an age, sex, and body mass index (BMI) matched community-based healthy controls (HCs) without knee OA in terms of the frequency of high-risk of osteoporotic fracture.

## 2. Materials and Methods

### 2.1. Study Design and Subjects

Our study was designed to be 1:1 matching between the knee OA and HCs. We defined knee OA using the American College of Rheumatology radiologic and clinical criteria for knee OA [15]. Status of radiologic knee OA was assessed using the Kellgren-Lawrence grade [16]. We retrospectively assessed 282 Korean patients with knee OA who visited five medical centers between November 2012 and November 2017. Subjects were excluded if they had confounding disorders such as rheumatoid arthritis, avascular necrosis, osteomyelitis, premature menopause, metabolic bone disease, malignancy within 5 years, high-impact trauma, or use of medications such as glucocorticoids and calcitonin within the last 3 months. After adjusting for the exclusion criteria, 245 patients with knee OA were selected. For the control group, we retrospectively assessed patients identified from the databases of two of the five medical centers that recruited patients with knee OA. Candidates were randomly selected using hospital registration numbers. A total of 1165 subjects aged ≥50 years were enrolled in the databases of health check-up centers. We excluded subjects who had self-reported OA or knee pain. We also excluded subjects who had radiologic defined knee OA higher than Kellgren-Lawrence grade 2 among those who had undergone knee X-ray. Overall, 991 subjects met the inclusion criteria before matching for age, sex, and BMI. Next, to control for major confounders of BMD, FRAX calculations, and frequency of high-risk osteoporotic fracture between the knee OA and HCs were matched for age, sex, and BMI. After matching for age, sex, and BMI; 478 subjects (239 knee OA patients and 239 HCs) were included in this study (Figure 1). The study was approved by the Institutional Review Board (IRB) of each hospital (AJIRB-MED-MDB-15-285, 3-32100191-AB-N-01, C2015163 (1621), DSMC2015-12-017-007, and 2015-09-026). Informed consents were waived by the IRBs.

### 2.2. BMD Evaluation

All subjects were evaluated for BMD using DXA (GE Lunar, Madison, WI, USA). The BMDs of the lumbar vertebrae (L1–4) and proximal femur were measured. On the basis of the WHO criteria, patients with normal BMD, osteopenia, and osteoporosis were classified according to BMD T-scores (standard deviation (SD) for a reference population) of ≥ −1, −1 > T-score > −2.5, and ≤ −2.5, respectively, for postmenopausal women or men aged ≥50 years. Patients with osteoporosis would be candidates for pharmacological intervention.

### 2.3. Osteoporotic Fracture Assessment Using the FRAX Calculation

FRAX uses data of clinical risk factors to estimate the 10-year probability of major osteoporotic and hip fractures. The FRAX values were calculated based on the Korean model (http://www.shef.ac.uk/FRAX/tool.aspx?country=25). The FRAX using BMD was calculated with femur BMD T-scores and the FRAX without BMD was calculated without femur BMD T-scores. According to the FRAX criteria, high-risk of osteoporotic fracture was defined as a 10-year probability of ≥ 20% for major osteoporotic fractures or ≥ 3% for hip fractures. Patients with a high risk of osteoporotic fractures would be candidates for pharmacological intervention.

### 2.4. Statistical Analysis

Continuous variables were expressed as mean ± SD. The *t*-test or Mann-Whitney test was used for the comparison of continuous variables in the prematched data of the knee OA and HCs. HC data were matched with the knee OA data (ratio of 1:1) by a statistician using greedy matching algorithms. The paired *t*-test or Wilcoxon signed rank test was used to compare differences between continuous variables in the matched data of the knee OA and HCs. McNemar’s test (for two categories) or test of marginal homogeneity (for more than three categories) was used to compare differences between categorical variables in the matched data of the knee OA and HCs. All statistical analyses were performed using SPSS version 21.0 (IBM Corp, Armonk, NY, USA). A *p* value of < 0.05 was considered statistically significant.

## 3. Results

The study population included 231 women (96.7%) and eight men (3.3%) in each group. All the female patients with knee OA and HCs were postmenopausal women. In the comparison of the BMD T-scores between the age, sex, and BMI-matched patients with knee OA and HCs, the patients with knee OA had lower proximal femur neck BMD T-scores than those with HCs (*p* = 0.036) (Table 1). However, the distributions of the BMD categories in patients with knee OA were similar to those in HCs (Table 2). Among the clinical risk factors assessed by FRAX, which were comparable between the two groups, previous fracture was the only difference identified between the knee OA and HCs (15.5% versus 0%, *p* < 0.001) (Table 3). The mean FRAX major osteoporotic fracture probabilities calculated with and without femur neck BMD were 6.9 ± 3.8% and 8 ± 3.6%, respectively, for the knee OA and 6.1 ± 2.8% and 6.8 ± 2.3%, respectively, for HCs; the FRAX calculations, regardless of the femur neck BMD T-scores, led to higher probabilities of major osteoporotic fracture in knee OA than in HCs (*p* = 0.000, and *p* < 0.001, respectively) (Figure 2A). The mean FRAX hip fracture probabilities calculated with and without femur neck BMD were 2.1 ± 2.4% and 3 ± 2.3%, respectively, for the knee OA and 1.7 ± 1.8% and 2.4 ± 1.6%, respectively, for HCs; The FRAX calculations, regardless of the femur neck BMD T-scores, led to higher probabilities of hip fracture in knee OA than in HCs (*p* = 0.006, and *p* < 0.001, respectively) (Figure 2B). Figure 3 shows the percentage of candidates eligible for pharmacological intervention between the two groups. Ninety-six of the 239 subjects (40.2%) were candidates for pharmacological intervention based on BMD values in the knee OA, while 87 of the 239 subjects (36.4%) were candidates in HCs. Candidate frequency for pharmacological intervention between the two groups was similar. FRAX calculations with the femur neck BMD indicated 22.7% and 18.9% of subjects would receive recommendations for osteoporosis treatment in the knee OA and HCs, respectively. This difference was also not significant. However, FRAX calculations without femur neck BMD indicated that 41.6% and 35.7% of subjects would receive recommendations for osteoporosis treatment in the knee OA and HCs, respectively. This difference was statistically significant (*p* = 0.008) (Figure 3). When comparing FRAX calculations between the patients with and without previous fractures among those with knee OA, the FRAX calculations, regardless of the femur neck BMD T-scores, were higher in the patients with previous fractures than in those without previous fractures (*p* < 0.0001) (Table 4). In addition, we compared the distributions of the BMD categories and FRAX probabilities in knee OA and HCs except for patients with previous fractures and male patients. The distributions of the BMD categories in the patients with knee OA were similar to those in the patients with HCs (Table 5). The FRAX calculations with the femur neck BMD were also similar between the two groups. However, the FRAX calculations without femur BMD were higher in the patients with knee OA than in those with HCs (*p* < 0.0001) (Table 6). In addition, when FRAX calculations without femur neck BMD were adjusted, an additional 10 patients with knee OA were recommended for osteoporosis treatment compared to HCs (*p* = 0.025) (Table 6). We investigated the distributions of the BMD categories between the two groups on the basis of the FRAX criteria in the patients with knee OA. In the patients classified as at high-risk of osteoporotic fracture using the FRAX calculations with femur neck BMD, the prevalence of osteoporosis was 54.0% (Table 7). In the patients classified as at high-risk of osteoporotic fracture using the FRAX calculations without femur neck BMD, the prevalence of osteoporosis was 88.9% (Table 8). The results indicated that the frequency of osteoporosis was higher in the patients classified as at high-risk of osteoporotic fracture than those not classified as at high-risk of osteoporotic fracture (Table 7 and Table 8).

## 4. Discussion

In our study, when using the WHO criteria based on BMD, the frequency of osteoporosis was 40.2% in patients with knee OA and 36.4% in HCs, which was not a statistically significant difference (Figure 3). This was similar to the prevalence of osteoporosis in the general female population in Korea [17]. Previous studies have confirmed that 20–29% of patients with advanced OA awaiting joint arthroplasty have occult osteoporosis [7,8,9]. A study in Korea also reported that the prevalence of osteoporosis in advanced knee OA patients was 31% [18]. However, the subjects of our study were not limited to advanced OA patients who were scheduled for arthroplasty as in the aforementioned study. We also found a high frequency of osteoporosis in patients with knee OA that was comparable to that in HCs, which differed from previous studies indicating an inverse relationship between osteoporosis and OA [1,2]. A cohort study reported no significant difference in BMD between patients surgically treated for hip or knee OA and a control group over five years. However, at the five-year follow up, OA was accompanied by changes in shape and a faster reduction of BMD [19]. This study supports our observation that the knee OA group did not have a significantly higher BMD than HCs. We confirmed that OA did not exert a protective effect on osteoporosis. In contrast to the WHO criteria, FRAX calculations regardless of the femur neck BMD T-scores led to higher probabilities of major osteoporotic fracture and hip fracture in the knee OA than in HCs (Figure 2). In addition, when FRAX calculations without the femur neck BMD were adjusted, 5.9% more patients in the knee OA would be recommend for osteoporosis treatment compared to HCs; 41.6% and 35.7% of subjects would receive recommendations for osteoporosis treatment in the knee OA and HCs (Figure 3). While the recommended proportion of patients is 5.9%, the number needed to treat for any clinical difference may be very large in clinical settings. We also indicated the classification of patients with knee OA who are at high-risk of osteoporotic fracture on the basis of FRAX selected patients with low BMD (Table 7 and Table 8). The FRAX calculations without femur neck BMD showed a higher detection rate of osteoporosis in the patients classified as at high-risk of osteoporotic fracture than the FRAX calculations with femur neck BMD. In our study, FRAX without BMD was more sensitive than FRAX with BMD in identifying patients with knee OA who were at high-risk of osteoporotic fracture. These results suggest that BMD may be a confounding factor that underestimates the frequency of high-risk osteoporotic fractures in patients with knee OA. In addition, a previous study demonstrated that the use of FRAX without BMD was comparable with that of FRAX with BMD [20]. Thus, according to the FRAX management algorithm [21], patients with knee OA classified as high risk using the FRAX without BMD may be offered treatment without BMD testing. In our study, of the 74 patients with knee OA classified as high risk using FRAX without BMD (Table 6), only 37 were classified as having osteoporosis. Further, 50% of patients with knee OA classified as high risk using the FRAX without BMD did not have osteoporotic BMD. This result suggests that FRAX without BMD may be applied in addition to BMD testing in patients with knee OA. The WHO criteria based on the value of the BMD T-scores ≤ −2.5 has been widely used as both a diagnostic and intervention threshold. Nevertheless, the value of the BMD T-scores showed high specificity but low selectivity, indicating discrepancy between BMD values and fragility fractures [10]. In previous studies, a number of fragility fractures occurred in individuals with BMD values above the osteoporosis threshold [11,22]. A prospective study indicated a higher incidence rate of fractures occurred in women with OA than in those without OA across all BMD groups [23]. This study also demonstrated that OA was a significant risk factor for any fracture in women with osteopenia or normal BMD, but not in osteoporotic women. This observation provides insight into non-osteoporotic fractures in OA patients and is consistent with the results of our study. In our study, 37 patients had previous fractures in knee OA, but none had previous fractures in HCs (Table 3). All 37 patients were female. Of the 37 patients with previous fractures, 30 (81.1%) are currently receiving antiosteoporosis treatment. We investigated the frequency of high-risk osteoporotic fracture in the 37 patients with previous fractures, comparing the WHO and FRAX criteria. On the basis of the WHO criteria, 22 patients (57.9%) were classified as having osteoporosis; 11 (29.7%), as having osteopenia; and four (10.8%), as having normal BMD. Fifteen patients (40.5%) with a T-score of ≥ −2.5 had previous fractures. The probability of developing a fracture using the FRAX calculation was higher in the patients with previous fractures than in those without previous fractures (Table 4). The FRAX calculations with femur neck BMD indicated that 17 (46.0%) of the 37 subjects would receive recommendations for osteoporosis treatment. The FRAX calculations without femur neck BMD indicated that 25 (67.6%) of the 37 subjects would receive recommendations for osteoporosis treatment. When the FRAX calculations without femur neck BMD were adjusted, three more patients (8.1%) from among the 37 patients with previous fracture would be recommended for osteoporosis treatment as compared with those when the WHO criteria were used. Because the goal of osteoporosis treatment is not to increase BMD but to prevent fracture, which ultimately determines disability and mortality, it may be necessary to compare each criterion used for osteoporosis treatment. A previous study compared FRAX without BMD and BMD alone for predicting osteoporosis treatment [24]. In that study, with advancing years, the difference in fracture probability between women with a T-score ≤ −2.5 and women of the same age without any risk factors decreased. The study showed the BMD criteria for intervention using a fixed T-score did not optimally target women at higher risk of fracture than age-matched individuals without any clinical risk factors, particularly among the elderly. Conversely, the probability of a major osteoporotic fracture in women with a previous fragility fracture increased with age, from 2.3% at the age of 40 years to 23% at the age of 90 years. Fracture probabilities based on the FRAX calculation were consistently higher in women with no clinical risk factors. In our study, the fracture probabilities derived from the FRAX calculation were higher in the knee OA than in HCs; these results may be due to the higher frequency of previous fractures in the knee OA. Therefore, when we consider that OA occurs mainly in elderly patients and that there is a higher fracture risk when previous fractures are taken into account, we may apply the FRAX calculations to patients with OA to assess fracture probabilities.

The high risk of osteoporotic fracture in OA patients may be associated with an increased fall tendency, possibly due to postural instability, quadriceps weakness, joint pain, and stiffness [3,5,25]. There are also studies that have reported bone loss due to OA increases the risk of osteoporotic fractures. A prospective study showed patients with radiographic hip and knee OA develop higher total hip bone loss over 2.6 years [26]. Another study indicated that radiographic hip OA was associated with an annual bone loss of 2% in men and 1.4% in women, despite the 3–8% higher BMD values when compared to control [27]. Other causes of increased fracture risk in OA patients could be inflammation. Inflammatory rheumatic diseases such as rheumatoid arthritis, systemic lupus erythematosus, and ankylosing spondylitis have been associated with elevated bone loss and increased fracture rates [28]. The contribution of inflammation to the development of osteoporosis is not limited to chronic inflammatory diseases, as the low grade inflammation of OA also contributes to the development of osteoporosis. Recent studies in the field of osteoimmunology have demonstrated that increased bone loss occurs not only in osteoporosis but also in the early stages of OA [29]. In our study, 144 (60.3%) complained of knee pain and 140 (58.6%) took analgesics, including nonsteroidal anti-inflammatory drugs, among the patients with knee OA. The distributions of the BMD categories in the patients with knee OA were similar to those in the patients with HC (Table 5). This suggests that the high frequency of high-risk osteoporotic fracture in patients with knee OA is due to increased fall tendency rather than to increased bone loss.

There are currently no data available comparing whether there is a difference in the development of osteoporotic fractures assessed by the WHO criteria and FRAX criteria in patients with knee OA, and whether these differences should affect osteoporosis treatment decisions. In our study, when adjusting for FRAX criteria but not for the WHO criteria, the frequency of high-risk osteoporotic fracture in patients with knee OA was higher than in HCs. Moreover, additional candidates for osteoporosis treatment were identified in patients with knee OA based on the FRAX criteria. Therefore, physicians may consider applying FRAX calculations in patients with knee OA to determine appropriate osteoporosis treatment. Whether the current treatments for osteoporotic fractures, which were targeted at individuals with low BMD, will be appropriate for patients with OA is yet to be determined. Additional prospective studies involving large populations are required to support the application of the FRAX calculation in patients with OA. We also suggest that further studies be conducted to develop a fracture risk assessment tool that includes OA as a risk factor, similar to the FRAX, which includes rheumatoid arthritis as a risk factor. Our study has some limitations. First, the number of knee OA patients included in the study was relatively small. In particular, the number of women (96.7%) was significantly higher than that of men. However, considering that female sex is a major risk factor for osteoporosis and knee OA [30,31], this study reflects the major concerns in a real world setting. Second, although we excluded subjects with self-reported OA, knee pain, or radiologic defined knee OA, there may still be a possibility of selection bias on HCs. Third, this study was restricted to knee OA, thus further studies are needed to determine whether the results of our study indicating a relationship between OA and osteoporotic fractures can be applied to other joints. Fourth, although patients with knee OA showed an increased risk of developing osteoporotic fractures, there are limitations in determining a clear causal relationship because of the cross-sectional study nature of our study.

## 5. Conclusions

This study demonstrated that the frequency of osteoporosis assessed by BMD was similar for patients with knee OA and HCs. In contrast, the probability of developing a fracture using the FRAX calculation was higher in the knee OA than in HCs. In addition, more candidates could be identified for osteoporosis treatment among knee OA patients considering the FRAX criteria. Thus, FRAX may have a clinical impact on treatment decisions aimed at reducing the development of osteoporotic fractures in patients with knee OA.

## Figures and Tables

**Figure 1 jcm-08-00918-f001:**
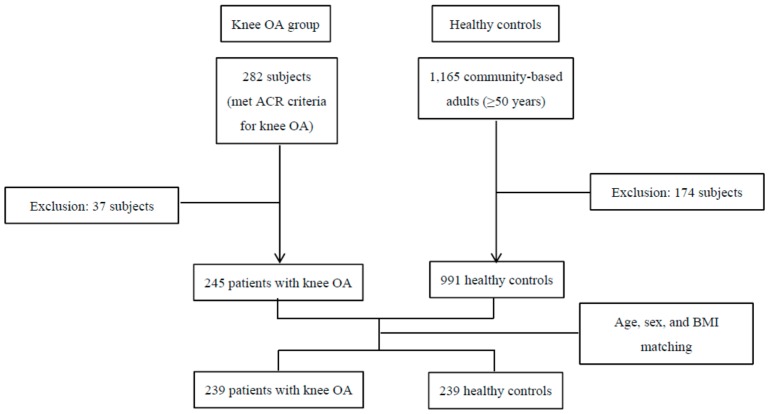
Subjects’ enrollment. OA, osteoarthritis; ACR, American College of Rheumatology; BMI, body mass index.

**Figure 2 jcm-08-00918-f002:**
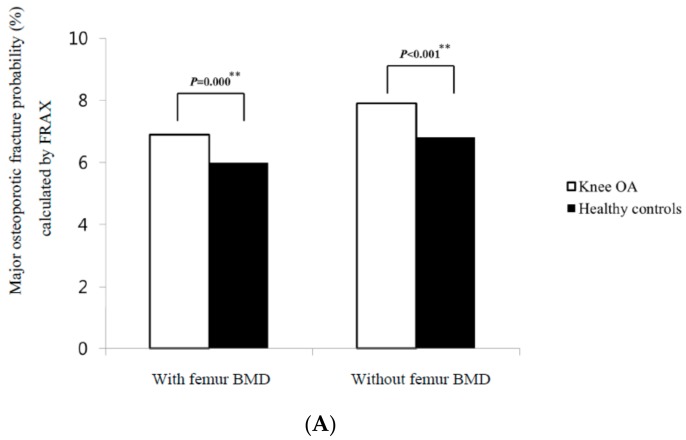
(**A**). Comparison of major osteoporotic fracture probabilities calculated by FRAX in the knee osteoarthritis and healthy controls. The results were expressed as mean ± SD. (**B**). Comparison of hip fracture probabilities calculated by FRAX in the knee osteoarthritis and healthy controls. The results were expressed as mean ± SD. * *p* < 0.05, ** *p* < 0.005, Statistical analysis was performed using paired *t*-test. FRAX, fracture risk assessment tool; SD, standard deviation; OA, osteoarthritis; BMD, bone mineral density.

**Figure 3 jcm-08-00918-f003:**
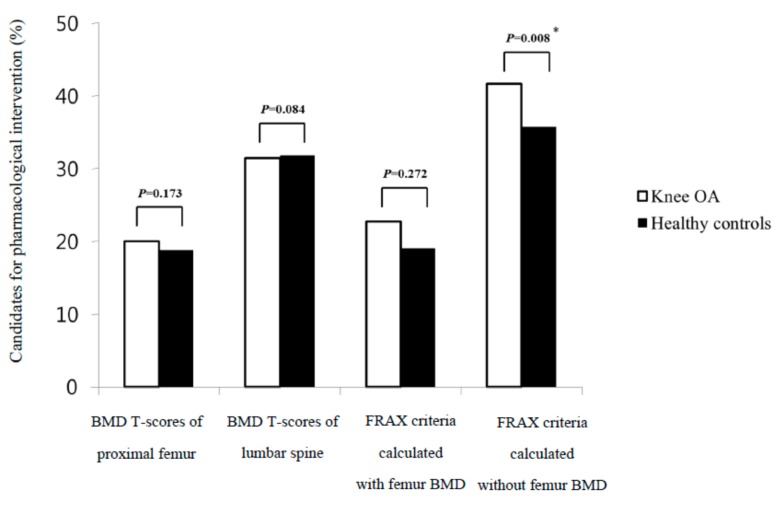
Candidates for pharmacological intervention by the WHO criteria and FRAX criteria. * *p* < 0.05. Statistical analysis was performed using McNemar’s test. WHO, World Health Organization; FRAX, Fracture risk assessment tool; OA, osteoarthritis; BMD, bone mineral density.

**Table 1 jcm-08-00918-t001:** Comparison of BMD T-scores between patients with knee osteoarthritis and healthy controls after age, sex, and BMI matching.

	Osteoarthritis *n* = 239	Healthy Controls *n* = 239	*p* Value
**Demographics**			
** Age (year), mean ± SD**	67.1 ± 9.3	67.1 ± 9.2	0.8244
** Sex (female), *n* (%)**	231 (96.7)	231 (96.7)	-
** Height (cm), mean ± SD**	153.1 ± 6.7	153.2 ± 6.5	0.7682
** Weight (kg), mean ± SD**	56.4 ± 9.3	56.7 ± 9.2	0.3427
** BMI (kg/m^2^), mean ± SD**	24 ± 3.5	24 ± 3.4	0.9676
**BMD T-scores, mean ± SD**			
** Proximal femur**			
** Neck**	−1.6 ± 1.0	−1.4 ± 1.1	0.036 *
** Femur total**	−1.2 ± 1.1	−1.2 ± 1.4	0.712
** Lumbar spine**	−1.8 ± 1.3	−1.8 ± 1.4	0.646

* *p* < 0.05. Statistical analysis was performed using paired *t*-test. BMD, bone mineral density; BMI, body mass index; SD, standard deviation.

**Table 2 jcm-08-00918-t002:** Comparison of distributions of the BMD categories between patients with knee osteoarthritis and healthy controls.

	Osteoarthritis *n* = 239	Healthy Controls *n* = 239	*p* Value
**Femur BMD T-scores, *n* (%)**			0.173
** Normal**	54 (22.6)	69 (28.9)	
** Osteopenia**	137 (57.3)	125 (52.3)	
** Osteoporosis**	48 (20.1)	45 (18.8)	
**Lumbar BMD T-scores, *n* (%)**			0.084
** Normal**	49 (20.8)	66 (28)	
** Osteopenia**	113 (47.9)	95 (40.3)	
** Osteoporosis**	74 (31.4)	75 (31.8)	

Statistical analysis was performed using the test of marginal homogeneity. BMD, bone mineral density.

**Table 3 jcm-08-00918-t003:** Comparison of FRAX clinical risk factors between patients with knee osteoarthritis and healthy controls.

	Osteoarthritis *n* = 239	Healthy Controls *n* = 239	*p* Value
**Age (year), mean ± SD**	67.1 ± 9.3	67.1 ± 9.2	0.8244
**BMI (kg/m²), mean ± SD**	24 ± 3.5	24 ± 3.4	0.9676
**Sex (female), *n* (%)**	231 (96.7)	231 (96.7)	-
**Previous fracture, *n* (%)**	37 (15.5)	0 (0)	<0.001 *
**Parent fractured hip, *n* (%)**	0 (0)	0 (0)	-
**Current smoking, *n* (%)**	1 (0.5)	3 (1.4)	0.317
**>3 units of alcohol per day, *n* (%)**	0 (0)	2 (0.9)	0.157

* *p* < 0.005. Statistical analysis was performed using McNemar’s test. FRAX, fracture risk assessment tool; SD, standard deviation; BMI, body mass index.

**Table 4 jcm-08-00918-t004:** Comparison of 10-year fracture probabilities calculated using the FRAX for the patients with or without previous fractures among the patients with knee osteoarthritis.

	Patients with Previous Fractures *n* = 37	Patients without Previous Fractures *n* = 202	*p* Value
**FRAX calculations, Mean ± SD**			
**Major osteoporotic fracture** ** probability with femur BMD**	10.4 ± 5.7	6.3 ± 2.9	<0.0001 *
**Hip fracture probability** ** with femur BMD**	4.0 ± 4.3	1.8 ± 1.7	<0.0001 *
**Major osteoporotic fracture** ** probability without femur BMD**	11.6 ± 4.2	7.3 ± 3.1	<0.0001 *
**Hip fracture probability** ** without femur BMD**	4.7 ± 2.5	2.7 ± 2.1	<0.0001 *

* *p* < 0.005. Statistical analysis was performed Mann-Whitney test. FRAX, fracture risk assessment tool; SD, standard deviation; BMD, bone mineral density.

**Table 5 jcm-08-00918-t005:** Comparison of the distributions of the BMD categories between the patients with knee osteoarthritis and the healthy controls except for patients with previous fractures and male patients.

	Osteoarthritis *n* = 194	Healthy Controls *n* = 194	*p* Value
**Femur BMD T-scores, *n* (%)**			0.461
** Normal**	46 (23.7)	55 (28.4)	
** Osteopenia**	114 (58.8)	105 (54.1)	
** Osteoporosis**	34 (17.5)	34 (17.5)	
**Lumbar BMD T-scores, *n* (%)**			0.053
** Normal**	39 (20.4)	56 (29.3)	
** Osteopenia**	95 (49.7)	77 (40.3)	
** Osteoporosis**	57 (29.8)	58 (30.4)	

Statistical analysis was performed using the test of marginal homogeneity. BMD, bone mineral density.

**Table 6 jcm-08-00918-t006:** Comparison of 10-year fracture probabilities calculated using the FRAX, and candidates for pharmacological intervention by the FRAX criteria in patients with knee osteoarthritis and healthy controls, except those with previous fractures and male patients.

	Osteoarthritis *n* = 194	Healthy Controls *n* = 194	*p* Value
**FRAX calculations, Mean ± SD**			
**Major osteoporotic fracture probability with femur BMD**	6.4 ± 2.9	6.0 ± 2.6	0.104
**Hip fracture probability with femur BMD**	1.8 ± 1.8	1.6± 1.6	0.090
**Major osteoporotic fracture probability without femur BMD**	7.5 ± 3.0	6.8 ± 2.2	<0.0001 **
**Hip fracture probability without femur BMD**	2.7 ± 2.1	2.3 ± 1.6	<0.0001 **
**Candidates for pharmacological intervention, *n* (%)**			
** FRAX with femur BMD**	37 (19.1)	34 (17.5)	0.691
** FRAX without femur BMD**	74 (38.1)	64 (33.0)	0.025 *

* *p* < 0.05, ** *p* < 0.005. Statistical analysis was performed using paired *t*-test and McNemar’s test. FRAX, fracture risk assessment tool; SD, standard deviation; BMD, bone mineral density.

**Table 7 jcm-08-00918-t007:** Distributions of the BMD categories between the two groups based on the FRAX calculations with femur BMD T-scores in the patients with knee osteoarthritis.

	Non High-Risk Group of Osteoporotic Fracture	High-Risk Group of Osteoporotic Fracture	*p* Value
**BMD T-scores, *n* (%)**			<0.001 *
** Normal**	24 (17.3)	3 (3.0)	
** Osteopenia**	73 (52.5)	43 (43.0)	
** Osteoporosis**	42 (30.2)	54 (54.0)	

* *p* < 0.005, Statistical analysis was performed using Fisher’s exact test. BMD, bone mineral density; FRAX, fracture risk assessment tool.

**Table 8 jcm-08-00918-t008:** Distributions of the BMD categories between the two groups based on the FRAX calculations without femur BMD T-scores in the patients with knee osteoarthritis.

	**Non high-risk group** **of osteoporotic fracture**	**High-risk group** **of osteoporotic fracture**	***p* Value **
**BMD T-scores, *n* (%)**			< 0.001 *
** Normal**	27 (14.6)	0 (0.0)	
** Osteopenia**	110 (59.5)	6 (11.1)	
** Osteoporosis**	48 (26.0)	48 (88.9)	

* *p* < 0.005. Statistical analysis was performed using Fisher’s exact test. BMD, bone mineral density; FRAX, fracture risk assessment tool.

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
