# Peer review of "Clinical Impact of the Fracture Risk Assessment Tool on the Treatment Decision for Osteoporosis in Patients with Knee Osteoarthritis: A Multicenter Comparative Study of the Fracture Risk Assessment Tool and World Health Organization Criteria"

_jcm, 2019, doi:10.3390/jcm8070918_

Reviewer 1 Report

Thank you for the revised manuscript. Although BMI reflects height and weight, and is practical in assessing obesity, two persons with totally different body image can have the same BMI. A person with 185 cm and 92 kg, and another with 158 cm with 67 kg have the same BMI, but totally different body image. Please, add height and weight into tables with BMI.

Author Response

Thank you for your comments. We seriously reviewed all your comments again. We marked the addition in the manuscript in red.

Thank you for the revised manuscript. Although BMI reflects height and weight, and is practical in assessing obesity, two persons with totally different body image can have the same BMI. A person with 185 cm and 92 kg, and another with 158 cm with 67 kg have the same BMI, but totally different body image. Please, add height and weight into tables with BMI.

Answer) We added height and weight into table 1. 

Reviewer 2 Report

Although the quality of the revised version improved, I still have some additional comments for authors to address.

My original comment 1:

First of all, I am not sure why WHO criteria and FRAX need to be compared to assess which modality better fit into OA population. WHO criteria is used to diagnose osteoporosis, FRAX with BMD is used to predict fracture in 10 years, although both modalities are used to identify patients at high risk of fractures for treatment intervention, they serve a different role in fracture prevention pipeline. If BMD shows osteoporosis, patient will be treated. If BMD within osteopenia range, FRAX will be applied with BMD included to calculate fracture probability in 10 years. BMD and FRAX will all be useful for OA population since they serve different clinical purposes.

Authors’s responses on comment 1:

OA is a significant risk factor for any fracture in women. A prospective study indicated a higher incidence rate of fractures occurred in women with OA than in those without OA across all BMD groups [2]. In our study, 37 patients had previous fractures in knee OA, but none had previous fractures in HC. Because the goal of osteoporosis treatment is not to increase BMD but to prevent fracture, which determines disability and mortality, it may be necessary to compare each criterion used for osteoporosis treatment. In our study, the distributions of the BMD categories in patients with knee OA were similar to those in HCs. In contrast, when adjusting for FRAX criteria, the frequency of high-risk osteoporotic fracture in patients with knee OA was higher than in HCs. In addition, additional candidates for osteoporosis treatment were identified in patients with knee OA based on the FRAX criteria. We further compared the FRAX probabilities in the knee OA and HCs except for patients with previous fractures and male patients. FRAX calculations without femur BMD were still higher in patients with knee OA than in HCs. Therefore, We suggest that antiosteoporosis treatment may be considered in accordance with FRAX management algorithm rather than BMD criteria in OA patients.

[2] Chan MY, Center JR, Eisman JA, Nguyen TV. Bone mineral density and association of osteoarthritis with fracture risk. Osteoarthritis Cartilage 2014;22:1251-8. PMID: 25042553

Additional comment:

It is no surprising to see that FRAX with BMD captured more patients at high risk of fracture in this study since patients with BMD<-2.5 were also included in the FRAX algorism, compared to that only osteopenia was included in the FRAX to predict fracture as clinically indicated. According to table 5, there were 34 OA patients without hx of fractures who had osteoporosis, were these 34 patients all captured by FRAX without BMD as high risk of fractures? If BMD on OA patients indicates osteoporosis, would you start treatment? If some patients with osteoporosis were not captured by FRAX without BMD as high-risk patients requiring treatment, I would say BMD still serve an important role in guiding treatment and preventing fractures, rather than as authors suggested, only run FRAX on OA patients without testing their BMD (even when they met BMD screening criteria such as aged > 65).    

My original comment 2:

This study may suffer severe selection bias on HC group. It is misleading to use “healthy” control since they are indeed non-OA group, they may have other risk factors or comorbidities, they are called healthy simply because they did not have documented or self-reported OA. Since the study objective is to compare FRAX calculated fracture probability between OA and HC groups, HC should be matching with OA by all important fracture risk factors such as history of previous fracture, it would be inadequate to control selection bias by only matching for age, sex and BMI. Table 3 supports this observation, 30(16%) of OA group had previous fracture vs. none from HC had previous fracture. Furthermore, whether those patients in OA group with previous fracture ever or currently on anti-osteoporosis treatment was not mentioned. These limitations would all potentially bias fracture probability comparison results between two groups, which could directly affect study findings and conclusion. 

Authors’s responses on comment 2:

We accepted your opinion to control selection bias and compared distribustions of the BMD categories and the FRAX probabilities in the knee osteoarthritis and healthy controls except for patients with previous fractures and male patients (Table 5-6). The FRAX calculations without femur BMD were still higher in the patients with knee OA than in those with HC (p<0.0001; Table 6). The relevant contents are described in red in the manuscript. “Of the 37 patients with previous fractures, 30 (81.1%) are currently receiving antiosteoporosis treatment.”- It was described in the manuscript (p.12)

Additional comment:

I would suggest authors to report what you described above in limitation paragraph to address possibility of selection bias.

My original comment 3:

Why McNemar’s test was used for comparing previous fracture, chi-square / fisher’s exact test should be used instead. McNemar’s test is used for assessing agreement between matched pairs of nominal data.

Authors’s responses on comment 3:

Table 3 shows the comparing whether there are FRAX clinical risk factors or not in the age, sex, and BMI matched data of the knee OA healthy controls. The McNemar’s test was used to compare the differences between categorical variables in the matched data of knee osteoarthritis and healthy controls.

Additional comment:

I don’t believe McNemar’s test was appropriately used and Chi-Square should be applied. The McNemar is not testing for independence, but consistency in responses across two variables on the same subject (i.e. response before and after treatment).  I would suggest to forward it to journal’s statistician for further review.

My original comment 4:

FRAX without BMD was not designed to predict fracture and has not been validated for this purpose. FRAX with BMD is supposed to be more sensitive than FRAX without BMD in identifying patients at high risk of fracture. However, it is intriguing to see the study findings suggest a paradox in OA population. FRAX without BMD appears to have higher sensitivity in predicting patients with high risk of fracture. I would encourage authors to focus on this findings in results and discussion (i.e. assessment in 38 patients with previous fractures, no table, not reported in Results, only briefly reported in Discussion, Line 269-279,). Not quite clear about what this finding is revealing, is it suggesting that patients with OA will more likely fall compared to HC since proportion of patients with osteoporosis based on femur and lumbar spine BMD in this study are comparable between OA and HC?

Authors’s responses on comment 4:

1) “In our study, FRAX without BMD was more sensitive than FRAX with BMD in identifying patients with knee OA who were at high-risk of osteoporotic fracture. These results suggest that BMD may be a confounding factor that underestimates the frequency of high-risk osteoporotic fractures in patients with knee OA. In addition, a previous study demonstrated that the use of FRAX without BMD was comparable with that of FRAX with BMD [20]. Thus, all patients with knee OA do not require a BMD test. According to the FRAX management algorithm [21], patients with knee OA classified as at high-risk using the FRAX without BMD may be offered treatment without BMD testing. Conversely, patients with knee OA classified as at low-risk using the FRAX without BMD may be denied treatment without BMD testing.“- It was described in the manuscript (p.11)

Additional reviewer’s comment:

Same question for authors: If patients with knee OA has BMD of <-2.5, would you treat patient? If not, why? If yes, it seems to be a bold statement that all patients with knee OA do not require a BMD test as authors suggested. To support the statement that patients with knee OA classified as at low-risk using the FRAX without BMD may be denied treatment without BMD testing, can authors confirm that none of those OA subjects with osteoporosis were labeled as low risk group by FRAX without BMD?

Author Response

Thank you for your comments. We seriously reviewed all your comments again. We marked the addition or change in the manuscript in red.

1)     Additional comment:

It is no surprising to see that FRAX with BMD captured more patients at high risk of fracture in this study since patients with BMD<-2.5 were also included in the FRAX algorism, compared to that only osteopenia was included in the FRAX to predict fracture as clinically indicated. According to table 5, there were 34 OA patients without hx of fractures who had osteoporosis, were these 34 patients all captured by FRAX without BMD as high risk of fractures? If BMD on OA patients indicates osteoporosis, would you start treatment? If some patients with osteoporosis were not captured by FRAX without BMD as high-risk patients requiring treatment, I would say BMD still serve an important role in guiding treatment and preventing fractures, rather than as authors suggested, only run FRAX on OA patients without testing their BMD (even when they met BMD screening criteria such as aged > 65).    

4) Additional reviewer’s comment:

Same question for authors: If patients with knee OA has BMD of <-2.5, would you treat patient? If not, why? If yes, it seems to be a bold statement that all patients with knee OA do not require a BMD test as authors suggested. To support the statement that patients with knee OA classified as at low-risk using the FRAX without BMD may be denied treatment without BMD testing, can authors confirm that none of those OA subjects with osteoporosis were labeled as low risk group by FRAX without BMD?

Answer)  We reviewed data again. Of the 194 healthy controls, 69 were classified as having osteoporosis. Of the 69 patients with osteoporosis, 35 were classified as high risk.of osteoporotic fracture on the basis of FRAX without BMD. Of the 194 in patients with knee OA, 70 were classified as having osteoporosis. Of the 70 patients with osteoporosis, 37 were classified as high risk.of osteoporotic fracture on the basis of FRAX without BMD. All patients with osteoporosis were not captured by FRAX without BMD as high risk of features. We agree with your opinion that BMD still serve an important role in guiding treatment and prevent fractures.

-  “all patients with knee OA do not require a BMD test” - This sentence was deleted.

-Thus, according to the FRAX management algorithm [21], patients with knee OA classified as high risk using the FRAX without BMD may be offered treatment without BMD testing. In our study, of the 74 patients with knee OA classified as high risk using FRAX without BMD (table 6), only 37 were classified as having osteoporosis. Further, 50% of patients with knee OA classified as high risk using the FRAX without BMD did not have osteoporotic BMD. This result suggests that FRAX without BMD may be applied in addition to BMD testing in patients with knee OA.  - The contents are changed as described . (p.11)

2)     Additional comment:

I would suggest authors to report what you described above in limitation paragraph to address possibility of selection bias.

Answer) The sentence was added in the manuscript (p.15); Second, although, we excluded subjects with self-reported OA, knee pain, or radiologic defined knee OA, there may still be a possibility of selection bias on HCs.

3)     Additional comment:

I don’t believe McNemar’s test was appropriately used and Chi-Square should be applied. The McNemar is not testing for independence, but consistency in responses across two variables on the same subject (i.e. response before and after treatment).  I would suggest to forward it to journal’s statistician for further review.

Answer) We consulted the statistician who had analyzed the data of this study on this issue. Table 3 shows the comparing whether there are FRAX clinical risk factors or not in the age, sex, and BMI matched data of the knee OA healthy controls. The McNemar’s test was used to compare the differences between categorical variables in the matched data of knee osteoarthritis and healthy controls.

- We present two related journals to prove that the McNemar’s test was used appropriately.

Journal 1) NEJM 1999;340:669-76 : “ A muticomponent intervention to prevent delirium in hospitalized older patients”

p.672 - related contents are indicated in yellow. 

Journal 2) NEJM 2015;373:328-38: “ Long-term outcomes of abdominal aortic aneurysm in the medicare population”

p.330 - related contents are indicated in yellow. 

Round  2

Reviewer 2 Report

I have no further comments to be addressed.